# Intratumor Microbiome in Neuroendocrine Neoplasms: A New Partner of Tumor Microenvironment? A Pilot Study

**DOI:** 10.3390/cells11040692

**Published:** 2022-02-16

**Authors:** Sara Massironi, Federica Facciotti, Federica Cavalcoli, Chiara Amoroso, Emanuele Rausa, Giovanni Centonze, Fulvia Milena Cribiù, Pietro Invernizzi, Massimo Milione

**Affiliations:** 1Division of Gastroenterology, Center for Autoimmune Liver Diseases, Ospedale San Gerardo, Department of Medicine and Surgery, University of Milano-Bicocca, 20900 Monza, Italy; pietro.invernizzi@unimib.it; 2Department of Biotechnology and Bioscience, University of Milano-Bicocca, 20126 Milan, Italy; federica.facciotti@unimib.it; 3Diagnostic and Therapeutic Endoscopy Unit, Fondazione IRCCS Istituto Nazionale Tumori, 20133 Milan, Italy; federica.cavalcoli@istitutotumori.mi.it; 4Department of Experimental Oncology, European Institute of Oncology IRCCS, 20141 Milan, Italy; chiara.amoroso@ieo.it; 5General, Emergency and Trauma Surgery Department, Papa Giovanni XXIII Hospital, 24121 Bergamo, Italy; erausa@asst-pg23.it; 6First Pathology Unit, Fondazione IRCCS Istituto Nazionale dei Tumori, 20133 Milan, Italy; giovanni.centonze@istitutotumori.mi.it (G.C.); massimo.milione@istitutotumori.mi.it (M.M.); 7Division of Pathology, Fondazione IRCCS Cá Granda Ospedale Maggiore Policlinico, 20122 Milan, Italy; shivania@hotmail.it

**Keywords:** neuroendocrine tumors, pancreatic neuroendocrine neoplasms, gut microbiota, bacterial infiltration, bacterial invasion, confocal fluorescent microscopy, fluorescent in situ hybridization

## Abstract

Neuroendocrine neoplasms (NENs) are rare neoplasms with heterogeneous clinical behavior. Alteration in human microbiota was reported in association with carcinogenesis in different solid tumors. However, few studies addressed the role of microbiota in NEN. We here aimed at evaluating the presence of bacterial infiltration in neuroendocrine tumoral tissue. To assess the presence of bacteria, 20 specimens from pancreatic NEN (pan-NEN) and 20 from intestinal NEN (I-NEN) were evaluated through Fluorescent In situ Hybridization and confocal microscopy. Demographic data, pre-operative investigations, operative findings, pathological diagnosis, follow-up, and survival data were evaluated. Among I-NEN, bacteria were detected in 15/20 (75%) specimens, with high variability in microbial distribution. In eight patients, a high infiltration of microorganisms was observed. Among pan-NEN, 18/20 (90%) showed microorganisms’ infiltration, with a homogeneous microbial distribution. Bacterial localization in pan-NEN was observed in the proximity of blood vessels. A higher bacterial infiltration in the tumoral specimen as compared with non-tumoral tissue was reported in 10/20 pan-NEN (50%). No significant differences were observed in mean bacterial count according to age, sex, ki67%, site, tumor stage. Mean bacterial count did not result to be a predictor of disease-specific survival. This preliminary study demonstrates the presence of a significant microbiota in the NEN microenvironment. Further research is needed to investigate the potential etiological or clinical role of microbiota in NEN.

## 1. Introduction

Neuroendocrine neoplasms (NENs) comprise a heterogeneous group of neoplasms arising from cells of the diffuse neuroendocrine system. Among them, those arising from the gastrointestinal tract and pancreas, i.e., the gastro-entero-pancreatic-NENs (GEP-NEN), are the most frequent [1]. Although rare, the worldwide incidence of neuroendocrine tumors is rising and ranges from 3.24/100,000 in Northern Europe to 6.98/100,000 in the U.S. [2,3]. NEN are classified according to the World Health Organization (WHO) based on tumor morphology and grading into well-differentiated (G1-G2-G3) neuroendocrine tumors and poorly differentiated neuroendocrine carcinoma (NEC G3) [4,5].

Remarkably, GEP-NENs present a marked heterogeneity in their clinical behavior depending on different sites of origin, grade, stage, growth rate, and the presence of the hormonal syndrome. Given this variable course of the disease, selection of the most suitable treatment remains challenging [6,7,8]. Several risk factors, which in part overlap with those predisposing to non-NEN solid cancers, have been recognized in GEP-NEN development. Interestingly, some of these risk factors, such as obesity-induced metabolic syndrome, and diabetes, have been recently linked as cooperating risk factors for digestive cancer via gut microbiota [9,10].

The gut microenvironment harbors a complex microbial ecosystem comprising approximately 3 × 1013 bacteria and other microorganisms that have co-evolved a mutualistic relationship with the host complementing its functions through dietary fibers fermentation, pathogen defense, and biosynthesis of vitamins and essential metabolites. This mutual interaction is further highlighted by its role in sustaining the maturation and functioning of the host’s immune system contributing to the host’s homeostasis [11]. Any imbalance in this delicate equilibrium may lead to an impaired microbiota condition, called dysbiosis, linked with several human pathologies, including cancer [12]. Therefore, the hypothesis that these factors may predispose to pre-neoplastic lesions through changes in the intestinal microbiota appears plausible [13,14]. Recently it has been proposed that chronic inflammation conditions driven by dysbiotic microbiota may be linked to the occurrence of gastrointestinal-NENs [15]. In line with that, type I gastric NENs are often associated with chronic atrophic gastritis, a complex condition that has been associated with long-standing infection of Helicobacter pylori, a Gram-negative bacterium whose lipopolysaccharides stimulate histamine release potentiating gastrin-driven DNA synthesis [2,16].

Albeit the physiological importance of bacteria within the intestine has been widely recognized through their effects on immune regulation, pathogen niche exclusion, and nutrition, none of these functions has been ascribed to the pancreas [17]. Several studies have detected the presence of microbiota within this organ, once thought sterile, in normal, nonpathological states [18,19]. However, results are conflicting, especially regarding microbial composition [18]. In addition, pancreatic acinar cells and islet cells are responsible for the secretion of antimicrobial peptides (AMPs), which represent ~10% of the proteins found in pancreatic juice that help in maintaining gut microbiota homeostasis and barrier function [17]. In recent years, a large number of studies have clarified that a variety of microorganisms (not only bacteria but also viruses, bacteria, and fungi) colonize pancreatic cancer tissues and are also closely related to the occurrence and development of pancreatic cancer, especially PDAC [20]. To date, the specific mechanisms through which the microbiome interreacts with the tumor microenvironment, including inflammatory induction, immune regulation, metabolism, and microenvironment changes, require further studies, also to identify new therapeutic targets as well as understanding whether the microbiome signature could affect early diagnosis, response to treatment and prognosis of several pancreatic neoplasms including neuroendocrine neoplasms.

Only a few studies are available in the literature on microbiota and GEP-NENs [2], and only one study evaluated microbial fecal composition using microscopic examination and fluorescence and in situ hybridization in 66 patients with NENs: a depletion of Fprau was observed in 67% of patients with midgut NENs, and fecal Enterobacteriaceae were significantly increased in intestinal NENs, similarly to what observed in Crohn’s Disease [21]. However, studies are very scanty and far from demonstrating a precise role of the microbiota in NEN. Moreover, a real analysis of microbiota on tumor tissue has not been performed, while most of the studies have focused on the association between microbiota and gastrointestinal adenocarcinomas. Since the relationship between the microbiota and GEP-NEN is still largely unknown, here we aimed at investigating the presence of bacteria within the pancreatic and intestinal NEN tumor tissue.

## 2. Materials and Methods

### 2.1. Study Design

This is a longitudinal study aimed at analyzing the presence of microbiota in 20 cases of pancreatic NEN (pan-NEN) and 20 cases of intestinal NEN (I-NEN). The study is a collaborative study between members of three research groups in three institutions: Fondazione IRCCS Ca’ Granda Ospedale Maggiore Policlinico, U.O.C. di Gastroenterologia ed Endoscopia, Fondazione IRCCS Istituto Nazionale dei Tumori, and European Institute of Oncology IRCCS.

To assess the presence or absence of bacteria in GEP-NEN tissues, formalin-fixed, paraffin-embedded (FFPE) tissues were evaluated through fluorescent in situ hybridization (FISH) by confocal microscopy.

Demographic data, pre-operative investigations, operative findings, pathological diagnosis (including tumor type, size, depth of invasion, degree of differentiation, resection margins, and lymph node involvement), grading, and staging have been evaluated. Follow-up and survival data have been collected from patient records.

The study protocol has been revised and approved by the ethical committees of the hospitals involved with ethics approval number INT 21/16.

### 2.2. Patients

Patients of age between 18 and 65 years who underwent surgical resection of pan-NEN or I-NEN were eligible for the study. Other inclusion criteria were the ability to provide full informed consent and Eastern Cooperative Oncology Group performance status >3. Exclusion criteria were: previous diagnosis of cancer, liver failure, renal insufficiency, decompensated heart failure, primary or secondary immunodeficiency, recent antibiotic treatment (1 month), autoimmune diseases, or relevant diet modification during the last month. Of the 20 patients with pan-NEN, 9 were female (45%), and the median age at diagnosis was 56 years (range 34–76). The median dimension at diagnosis was 2.75 mm (range 0.8–6 mm). In 3 cases (15%), the neoplasm was located at the head of the pancreas, in 4 cases (20%) at the body, and in the remaining 13 (65%) cases at the pancreatic tail. Most patients, 13 (65%), had a well-differentiated G1 neoplasm, 5 (25%) had a well-differentiated G2 neoplasm, and 2 (10%) presented with G3 NENs. With regards to stage six were at stage I (30%), five at stage II (25%), two at stage III (10%), and seven at stage IV (35%).

Regarding the 20 patients with I-NENs, 10 were female (50%), the median age at diagnosis was 62 years (range 33–77). The median dimension at diagnosis was 1.6 mm (range 0.8–5 mm). Overall, 15 patients (75%) had a well-differentiated G1 neoplasm, 4 (20%) had a well-differentiated G2 neoplasm, and 1 (5%) presented with G3 NENs. As regards stage, three patients (15%) were at stage II, six (30%) were at stage III, and 11 patients (55%) were at stage IV (Table 1).

### 2.3. Fluorescent In Situ Hybridization (FISH)

Formalin-fixed paraffin-embedded tissues, obtained from patients at IRCCS Istituto Nazionale dei Tumori (Milan, Italy), were sectioned to 5 μm thickness. Sections were deparaffinized with xylene and dehydrated in alcohol. The probes (EUB1, EUB2, and EUB3) used were designed to specifically target different regions of the 16S rRNA. All the probes were manufactured by Sigma-Aldrich (Milan, Italy) and labeled with Alexa488. Probes were applied to slides at a concentration of 5 ng/μL in prewarmed hybridization buffer (0.9 M NaCl, 20 mM Tris, pH 7.4, and 0.01% SDS). The slides were incubated overnight at 50 °C in a humid chamber and washed at 48 °C in prewarmed washing buffer (0.9 M NaCl and 20 mM Tris, pH 7.4). To avoid the strong autofluorescence in tissue, the Vector^®^ TrueVIEW^®^ autofluorescence quenching kit (Vector Laboratories, Burlingame, CA, USA) was used. The slides were counterstained with DAPI. The probes sequences are listed in Table 2.

### 2.4. Confocal Microscopy

Confocal images were acquired through an SP8 confocal microscope equipped with an HCX PL APO CS2 63X/1.40 oil immersion objective, 405, 488, 638 nm solid-state lasers, and 3 photomultiplier detectors (Leica Microsystem). The choice of the acquisition fields was made by the operator. Indeed, fields inside the section and not at the periphery were chosen to be sure that the detected bacteria were intra-tissue and were not coming from contamination. For each patient, 5 sections were acquired for pancreatic specimens, whereas 2–3 sections were for intestinal specimens. In addition, for pan-NEN, 5 sections from non-tumoral pancreatic tissue of the same patient were analyzed and compared with pan-NEN specimens. The difference in total bacterial abundance between the paired tumoral and tumor-free tissue (N bacteria Tumfree-N bacteriaTum) was calculated, and patients were subdivided into two groups: Group A, characterized by a statistically significantly higher bacterial abundance within the tumoral tissue, and Group B, by a lower bacterial abundance.

The DAPI-labeled nuclei were exited with the 405 nm laser line, and the emitted fluorescence was acquired between 415 and 480 nm. The A488-labeled bacteria were excited with the 488 nm laser line, and the emitted fluorescence was acquired between 495 and 540 nm. The Cy5-labeled non-bacteria population was excited with the 638 nm laser line, and the emitted fluorescence was acquired between 645 and 750 nm. Z-stacks of the tissue sections were acquired with a voxel size of 90 × 90 × 500 nm. The maximum intensity projections obtained from the acquired Z-stacks were analyzed with the ImageJ-based Fiji program (http://fiji.sc/Fiji accessed on 11 July 2020) in order to optimize the brightness and contrast settings in each picture. To quantify the number of bacteria in each field of view, only the bacteria showing the A488 signal (green channel, EUB1-2-3 population) and not showing the Cy5 signal (red channel, NON-EUB population) were counted.

### 2.5. Statistical Analysis

The results are given as median values and ranges unless otherwise stated.

Differences between variables were assessed by using a Mann–Whitney test, and a *p*-value < 0.001 was regarded as statistically significant.

The relationships between variables were assessed by logistic regression analysis and keeping the presence of either mean bacterial count or sum bacterial count as the dependent variables. Subgroup analyses according to age, sex, Ki67% index, TNM staging, somatostatin analogs therapy have been performed.

Disease-specific survival (DSS) was calculated from the date of NEN diagnosis to the patient’s death or the end of data collection. The univariate–multivariate Cox regression model was used to analyze the possible association between some covariates (mean bacterial count, sum bacterial count, age, sex, grading, staging) and the risk of death.

The analyses were carried out by software Graph Pad Prism version 6.00 (GraphPad Software, San Diego, CA, USA) and MedCalc version 17.9.5 (MedCalc Software bvba, Ostend, Belgium).

## 3. Results

### 3.1. Bacteria Colonize I-NEN and pan-NEN Tissues

To evaluate the infiltration of bacteria within tumoral tissues, FISH analyses with eubacterial probes were performed on PFFE specimens of intestinal and pancreatic NEN (Figure 1A). A total of 15 patients with I-NEN out of 20 (75%) showed bacteria infiltration inside tumoral tissues (Figure 1B). To note, interindividual variability was reported among the positive specimens concerning the microbial distribution, as shown by the cumulative plot in Figure 1C,D. Eight patients were characterized by an elevated infiltration of microorganisms. Conversely, in the remaining seven patients’ positive tissues, the number of bacteria present ranged between 1 and 38 bacteria for each acquired section (Figure 1C,D).

Interestingly, also pan-NEN tissues showed bacterial infiltration. Among the 20 pan-NEN specimens analyzed, 18 (90%) manifested countable microorganisms tissue infiltration (Figure 2A). Microbial distribution within pan-NEN tissues was more homogeneous than that of intestinal NEN patient’s derived specimens. Although the number of bacteria varied from section to section, microorganisms could be counted in all the pan-NENs specimens (Figure 2B).

### 3.2. Bacteria Infiltrate pan-NEN Tissues

To evaluate if bacterial infiltration specifically associates with tumoral tissues, paired tumoral and tumor-free tissues were evaluated by FISH analyses with eubacteria probes for each pan-NEN patient (Figure 3A). Interestingly, when the difference in total bacterial abundance between the paired tumoral and tumor-free tissue (N bacteria Tumfree-N bacteriaTum) was calculated, patients were subdivided into two groups (Figure 3B). Group A was characterized by a statistically significantly higher bacterial abundance within the tumoral tissue, while Group B by a lower bacterial abundance (Figure 3C).

### 3.3. Clinical Variables

At logistic regression, no statistically significant differences were observed in mean bacterial count according to patients’ age (*p* = 0.6241), sex (*p* = 0.7767), ki67% index (*p* = 0.4343), site (*p* = 0.8804) tumor stage (*p* = 0.1423), and somatostatin analogs therapy (*p* = 0.3352).

Finally, at Cox univariate analysis, mean bacterial count did not result as a predictor of disease-specific survival (DSS) (*p* = 0.1532).

## 4. Discussion

In recent years, several studies have reported the central role of the gut microbiota as a key determinant of numerous pathologic conditions, including cancer [12]. However, while most of the studies have focused on the association between microbiota and gastrointestinal adenocarcinomas, very little is known about GEP-NENs.

To our knowledge, we here demonstrated, for the first time, that both I-NEN and pan-NEN tissues are colonized by bacteria. Moreover, even though some studies [18,19] indicate the existence of a pancreatic microbiota in normal and in nonpathological states, this study clearly showed a higher bacterial infiltration in pancreatic tumors compared to non-tumoral pancreatic tissue. The fact that bacterial infiltrate in non-tumoral-pancreatic tissues resulted statistically lower than in tumoral specimens reinforces the hypothesis of a potential microbial translocation, even prior to tumor formation, specifically in cases of pan-NENs. This phenomenon could be explained by the fact that microbiota might gain access to the pancreatic parenchyma during carcinogenesis. However, to date, it is still unclear how bacteria translocate to the pancreas once having surmounted the intestinal barrier. In particular, the pancreas might be a privileged organ whereby pancreatic ductal epithelial tight junctions and antimicrobial peptide secretion act as a barrier function to prevent bacteria translocation and organ colonization [22,23]. Disturbances in these local defense mechanisms or an impaired antimicrobial peptide secretion could make the pancreas more susceptible to opportunistic bacterial colonization.

The physiological importance of bacteria within the intestine has been widely recognized through their effects on immune regulation, pathogen niche exclusion, and nutrition. However, none of these functions has been ascribed to the pancreas. The intimate relationship of the pancreas to the gastrointestinal tract raised the question of whether the intestinal microbiota or even an intrinsic pancreatic microbiota might impart similar homeostatic properties to this organ, as described in a recent review of Thomas and Jobin [17]. Few studies have detected the presence of a pancreatic microbiota, an organ once thought sterile, in normal, nonpathological states. For instance, Li S et al. analyzed the microbial constituents in the pancreatic cyst fluids, where *Bacteroides*, *Escherichia/Shigella*, and *Acidaminococcus* were predominant [24]. Therefore, the results are discordant. Additionally, in the literature, there are no data available regarding the presence of a microbial signature in pan-NENs patients. At the DDW 2021, the group of Nicholas Chia reported the signature profiling by 16S sequencing of snap-frozen (not FFPE sections) pan-NENs, PDAC, and acinar carcinomas, revealing that, irrespective of location, tumors were dominated by *Bacteroidetes*, *Firmicutes*, *Proteobacteria*, and *Actinobacteria*. Unfortunately, the information in this regard is very few and uncertain.

Several routes of pancreatic bacterial translocation have been proposed, including: (i) hematogenous seeding via the circulation, (ii) ascending infection from the duodenum via the main pancreatic duct, (iii) migration via the lymphatics, and (iv) seeding from the portal vein and the liver via the biliary duct system [25]. Interestingly, we observed that bacterial localized preferentially in the proximity of blood vessels, although we did not perform simultaneous staining of bacteria and vascular markers such as CD31. However, this finding may support the hypothesis of a microbial translocation into the impaired organ through portal circulation.

Although we reported high interindividual variability in the microbial abundance distribution within patients, we did not observe any significant differences in the amount of bacterial infiltration with regards to patients’ age, ki-67% index, or tumor stage, as well as no association between mean bacterial count and DSS. Somatostatin analogs did not influence bacterial count or bacterial groups, as already hypothesized by others [26]. We do acknowledge that this study has some limitations, including the low number of specimens evaluated and the lack of detailed taxonomic composition of the infiltrate.

However, this study documents for the first time that microbiota may importantly contribute to the tumor microenvironment in GEP-NEN, even if it remains to be elucidated whether this can play a pathogenetic, clinical, or prognostic role, as described for other intestinal [27,28] and pancreatic cancers [29,30,31].

In our study, FISH was performed to specifically target different regions of the bacterial 16S rRNA gene, which is usually fragmented in FFPE specimens. Indeed, bacterial genetic material can be damaged by the fixing process itself, along with the length of exposure to formalin, pH of formalin, and sample storage time. However, to avoid the detection of microbial contamination of pathological tissues, confocal images were acquired inside the section and not at the periphery. Thus, we were confident that the detected bacteria were intra-tissue and were not coming from secondary microbial colonization.

Moreover, since we detected bacterial DNA, speculations about their metabolism and their participation in the pathological (neoplastic) process cannot be assumed with the current analyses.

This preliminary study further suggests the possible existence of a crosstalk between intratumoral infiltrating bacteria and anti-tumor immunity, leading to taking into consideration the exciting implications of implementing therapeutic strategies targeting the microbiota-immune system axis even in this kind of tumor.

In conclusion, this study demonstrates the presence of a significative bacterial infiltrate as a part of the neuroendocrine tumor microenvironment, especially pan-NEN, giving the first hints for a potentially active role for the microbiota in the development and progression of neuroendocrine cancers of the gastroenteropancreatic tract. Further research is needed to obtain a deeper knowledge of the potential etiological or clinical role of the microbiota in the GEP-NEN. Although still at the beginning, this study provides a rationale for future studies aimed at diving deeper into the elucidation of possible microbial signatures to predict response to certain therapies or better disease outcomes.

## Figures and Tables

**Figure 1 cells-11-00692-f001:**
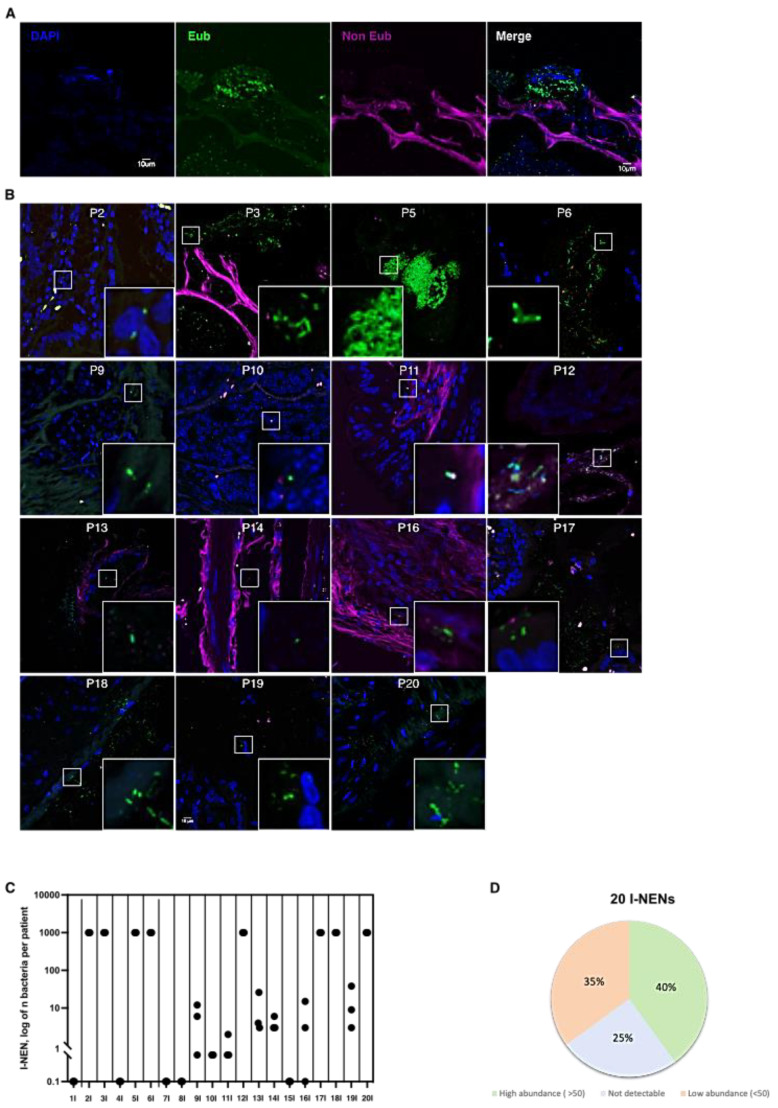
Bacterial count in I-NEN patients. (**A**) Confocal microscopy images of single optical sections showing the DAPI, Eub (Alexa 488), non-Eub (Cyanine-5) fluorescence signals that are merged together. (**B**) Representative confocal microscopy images of the bacterial signal observed within I-NEN patients (N= 15). Scale bar = 10 µm. (**C**) Dot plot representing the log10 normalized bacterial counts per acquired section (N = 3) by FISH in I-NEN specimens (N= 20). (**D**) Distribution of bacterial occurrence in I-NEN patients (N = 20).

**Figure 2 cells-11-00692-f002:**
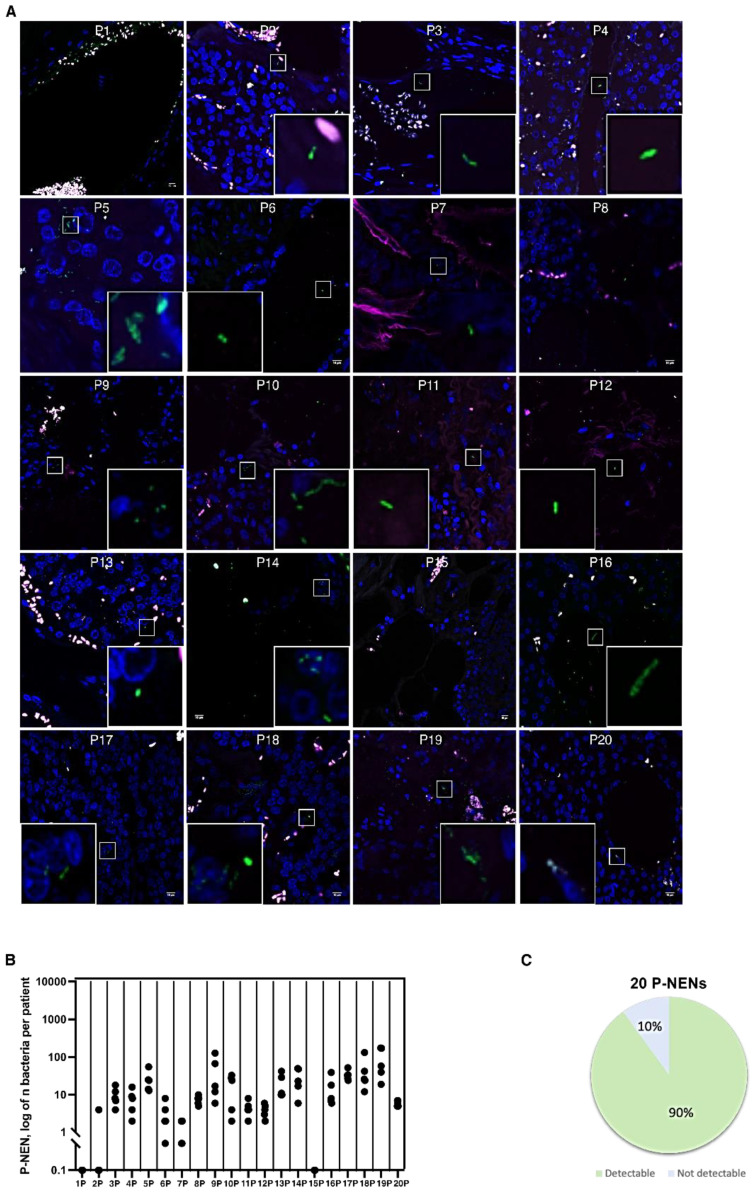
Bacterial count in pan-NEN patients. (**A**) Representative confocal microscopy images of the bacterial signal observed within pan-NEN patients (N = 18). Scale bar = 10 µm. (**B**) Dot plot representing the log10 normalized bacterial counts per acquired section (N = 5) by FISH in pan-NEN specimens (N = 20). (**C**) Distribution of bacterial occurrence in pan-NEN patients (N = 20).

**Figure 3 cells-11-00692-f003:**
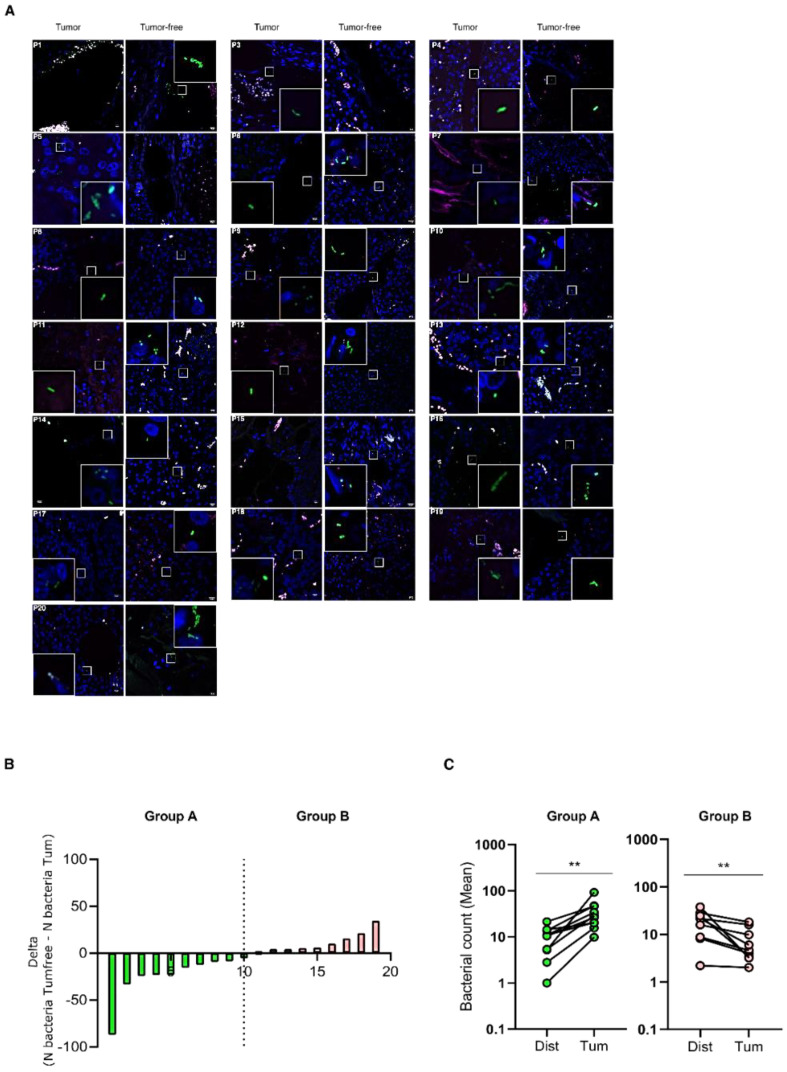
Comparison between bacterial infiltration in paired tumoral and tumor-free pan-NEN tissue. (**A**) Representative confocal microscopy images of the bacterial signal observed within paired tumoral and tumor-free pan-NEN patients (N = 18). Scale bar = 10 µm. (**B**) Bar plot representing the difference between bacteria abundance within paired tumoral and tumor-free tissue (N bacteria Tumfree-N bacteriaTum) (N = 19). (**C**) Comparison between the mean of bacterial count within paired tumoral and tumor-free tissue of Group A and B. Wilcoxon test was used to assess statistical significance. ** *p*-value < 0.05.

**Table 1 cells-11-00692-t001:** Demographic and clinical characteristics of the NEN patients.

Characteristics	Pan-NEN (N=20)	I-NEN (N20)
Age in years, median (range)	56 (34–76)	62 (33–77)
Sex, M/F	11/9	10/10
Dimension mm, median (range)	2.75 (0.8–6)	1.6 (0.8–5)
Ki-67%, median (range)	6.52 (0.1–45)	3.4 (0.1–28)
Grading, N (%):		
G1	13 (65)	15 (75)
G2	5 (25)	4 (20)
G3	2 (10)	1 (5)
Treatment, N (%)		
SSAs therapy	9 (45)	11 (55)
Chemotherapy	3 (15)	3 (15)
Target therapy	3 (15)	2 (10)
Radio ligand theraphy	2 (10)	2 (10)

**Table 2 cells-11-00692-t002:** Sequences of probes used for FISH analysis.

Probes	Sequences	Vendor
EUB338-1	5′(A488)-GCTGCCTCCCGTAGGA	SIGMA
EUB338-2	5′(A488)-GCAGCCACCCGTAGGTG	SIGMA
EUB338-3	5′(A488)-GCTGCCACCCGTAGGTG	SIGMA
Non338	5′(Cyanine5)-CGACGGAGGGCATCCTCA	SIGMA

## Data Availability

The data that support the findings of this study are available from the corresponding author.

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
