# Peer review of "Intratumor Microbiome in Neuroendocrine Neoplasms: A New Partner of Tumor Microenvironment? A Pilot Study"

_cells, 2022, doi:10.3390/cells11040692_

Round 1

Reviewer 1 Report

Dear Dr Massironi et al., 

Thank you for the manuscript. It is well-written, but I have some major comments. 

  1. I understand the connection between gut microbiota and intestinal tumors. However, it is not as clear with respect to pancreatic NET. Could you explain the connection to pancreatic patophysiology more thoroughly in the Introduction?
  2. I miss a description of how dangerous "bacterial infiltration" is. How many healthy people have bacterial infiltration? 
  3. Could you provide more details on the patients included? E.g. the Ki-67 index, treatments (SSA, PRRT etc.).
  4. Would the study benefit from comparison with a control group? It would allow you to use more advanced statistics.

Minor. 

Check the legend to Table 2.

Author Response

Thank you for the manuscript. It is well-written, but I have some major comments. 

I understand the connection between gut microbiota and intestinal tumors. However, it is not as clear with respect to pancreatic NET. Could you explain the connection to pancreatic pathophysiology more thoroughly in the Introduction?

In recent years, a large number of studies have clarified that a variety of microorganisms (not only bacteria but also viruses, bacteria, and fungi) colonize pancreatic cancer tissues and are also closely related to the occurrence and development of pancreatic cancer, especially PDAC.  As requested by the reviewer, this information has been added in the introduction (page 2, line 79-87), with proper citation.

I miss a description of how dangerous "bacterial infiltration" is. How many healthy people have bacterial infiltration?

That’s a very interesting point, unfortunately without a real and certain answer. The physiological importance of bacteria within the intestine has been widely recognized through their effects on immune regulation, pathogen niche exclusion, and nutrition. However, none of these functions has been ascribed to the pancreas. The intimate relationship of the pancreas to the gastrointestinal tract raised the question of whether the intestinal microbiota or even an intrinsic pancreatic microbiota, might impart similar homeostatic properties to this organ, as described in a recent review of Thomas and Jobin (10.1038/s41575-019-0242-7).  Few studies have detected the presence of a pancreatic microbiota, an organ once thought sterile, in normal, non-pathological states. For instance, Li S, et al. analyzed the microbial constituents in the pancreatic cyst fluids, where Bacteroides, Escherichia/Shigella, and Acidaminococcus were predominant (10.1186/s40168-017-0363-6). Therefore, results are discordant. Additionally, in literature, there are no data available regarding the presence of a microbial signature in Pan-NENs patients. At the DDW 2021, the group of Nicholas Chia reported the signature profiling by 16S sequencing of snap-frozen (not FFPE sections) pan-NENs, PDAC, and acinar carcinomas, revealing that, irrespective of location, tumors were dominated by Bacteroidetes, Firmicutes, Proteobacteria and Actinobacteria. Unfortunately the information in this regard is very few and uncertain.

A comment in the discussion section has been added (page 9, lines 266-281).

Could you provide more details on the patients included? E.g. the Ki-67 index, treatments (SSA, PRRT etc.).

As required, more details have been included in table 1 (pages 3 and 4)

Would the study benefit from comparison with a control group? It would allow you to use more advanced statistics.

The reviewer raises a very interesting point. However, it’s very difficult to have normal human pancreatic tissue to analyze. In our study, we compared the neoplastic tissue with the adjacent non-tumoral tissue and we interestingly observed that bacterial infiltrate in non-tumoral-pancreatic tissues resulted statistically lower than in tumoral specimens, reinforcing the hypothesis of a potential microbial translocation, even prior to tumor formation specifically in cases of pan-NENs (see “results” page 8, line 223-230 and “discussion”, page 9, lines 251-260) .

Minor.

Check the legend to Table 2.

As properly suggested, the legend of Table 2 has been corrected

Reviewer 2 Report

The idea of the potential participation of microbiota bacteria in the oncological process of pancreas is not new, it is actively discussed in the literature.  The article is an original study, new data were obtained using the Fish - method, which is certainly relevant. The study protocol has been revised and approved by the local ethics committees

But, a number of questions of paramount importance remain unclear:

  • are these microorganisms alive; to which group do they belong? Or remnants (fragments) of the genetic material of destroyed bacteria?
  • are they capable of active metabolism and participation in the pathological (neoplastic) process?
  • or are the authors' findings a reflection of the process of secondary microbial colonization of pathological tissue, just inactive «garbage»?

Thus, the authors rightly conclude that further research is needed to gain deeper knowledge about the potential etiological or clinical role of microbiota in neuroendocrine tumors.

It is necessary to carefully check and make a number of corrections (table names, figure captions).

For example:

Table 2.   (line 143)  This is a table. Tables should be placed in the main text near to the first time they are cited.  There is no name at all

Figure 3. The designations Group A and Group B (line 224) appear in Fig 3 and in the caption for the first time. What does it mean?   It is necessary to give a description of these groups in 2. Materials and Methods, section 2.2 Patients (line 108)

Author Response

The idea of the potential participation of microbiota bacteria in the oncological process of pancreas is not new, it is actively discussed in the literature.  The article is an original study, new data were obtained using the Fish - method, which is certainly relevant. The study protocol has been revised and approved by the local ethics committees

But, a number of questions of paramount importance remain unclear:

Are these microorganisms alive; to which group do they belong? Or remnants (fragments) of the genetic material of destroyed bacteria?

Are they capable of active metabolism and participation in the pathological (neoplastic) process?

Or are the authors' findings a reflection of the process of secondary microbial colonization of pathological tissue, just inactive «garbage»?

Thus, the authors rightly conclude that further research is needed to gain deeper knowledge about the potential etiological or clinical role of microbiota in neuroendocrine tumors.

We thank the reviewer for the positive comments.

FISH was perfomed to specifically target different regions of the bacterial 16S rRNA gene, which is usually fragmented in FFPE specimens. Indeed, bacterial genetic material can be damaged by the fixing process itself along with the length of exposure to formalin, pH of formalin and sample storage time. However, to avoid the detection of microbial contamination of pathological tissues, confocal images were acquired inside the section and not at the periphery. Thus, we were confident that the detected bacteria were intra-tissue and were not coming from a secondary microbial colonization.

Moreover, since we detected bacterial DNA, speculations about their metabolism and their participation in the pathological (neoplastic) process can not be assumed with the current analyses.

These comments have been included in the “discussion” section, page 10, lines 301-310.

Concerning the bacterial classification question, one could either sequence the extracted bacterial DNA from the FFPE sections or use bacterial taxa-specific targeted probes, which might be hypothesis driven or guided by data of metataxonomic analysis. Given the complexity of performing 16s sequencing on FFPE tissues, in this pilot study we aimed at evaluating the presence of bacteria in pan-NEN tissues. Indeed, to target the bacterial DNA of higher taxonomic levels of most of bacteria, we chose eubacterial probes (EUB338-1, EUB338-2, EUB338-3), which are not suitable to identify bacterial taxa.

Therefore, the use of targeted probes for specific bacterial genera and metataxonomic analysis of the patients’ intratumoral microbiota will help in highlighting the potential bacterial role in the development and progression of neuroendocrine cancers of the gastroenteropancreatic tract. In the current study, we did not perform this type of analysis, which will be hopefully subject of a next more detailed study.

It is necessary to carefully check and make a number of corrections (table names, figure captions).

For example:

Table 2.   (line 143)  This is a table. Tables should be placed in the main text near to the first time they are cited.  There is no name at all

As properly suggested, the legend of Table 2 has been corrected

Figure 3. The designations Group A and Group B (line 224) appear in Fig 3 and in the caption for the first time. What does it mean?   It is necessary to give a description of these groups in 2. Materials and Methods, section 2.2 Patients (line 108)

We thank the reviewer for his observations a description of Group A and B has been provided in Material and Methods section (page 4, lines 161-164).

Round 2

Reviewer 1 Report

Thank you for the comments, nothing further.